# Complications of Robotic Video-Assisted Thoracoscopic Surgery Compared to Open Thoracotomy for Resectable Non-Small Cell Lung Cancer

**DOI:** 10.3390/jpm12081311

**Published:** 2022-08-12

**Authors:** Oscar Zhang, Robert Alzul, Matheus Carelli, Franca Melfi, David Tian, Christopher Cao

**Affiliations:** 1School of Clinical Medicine, UNSW Medicine & Health, St. Vincent’s Healthcare Clinical Campus, Faculty of Medicine and Health, UNSW Sydney, Sydney, NSW 2010, Australia; 2Department of Cardiothoracic Surgery, Royal Prince Alfred Hospital, Sydney University, Sydney, NSW 2050, Australia; 3Robotic Multispecialty Center for Surgery Robotic, Minimally Invasive Thoracic Surgery, University of Pisa, 56124 Pisa, Italy; 4Department of Anaesthesia and Perioperative Medicine, Westmead Hospital, Westmead, NSW 2145, Australia; 5Chris O’Brien Lifehouse Hospital, Sydney, NSW 2050, Australia; 6Baird Institute for Applied Heart and Lung Surgical Research, Sydney, NSW 2042, Australia

**Keywords:** robotic-assisted surgery, minimally invasive surgery, lung cancer, lobectomy, meta-analysis, systematic review

## Abstract

(1) Background: Conventional open thoracotomy has been the accepted surgical treatment for resectable non-small cell lung cancer. However, newer, minimally invasive approaches, such as robotic surgery, have demonstrated similar safety and efficacy with potentially superior peri-operative outcomes. The present study aimed to quantitatively assess these outcomes through a meta-analysis. (2) Methods: A systematic review was performed using electronic databases to identify all of the relevant studies that compared robotic surgery with open thoracotomy for non-small cell lung cancer. Pooled data on the peri-operative outcomes were then meta-analyzed. (3) Results: Twenty-two studies involving 12,061 patients who underwent robotic lung resection and 92,411 patients who underwent open thoracotomy were included for analysis. Mortality rates and length of hospital stay were significantly lower in patients who underwent robotic resection. Compared to open thoracotomy, robotic surgery was also associated with significantly lower rates of overall complications, including atrial arrhythmia, post-operative blood transfusions, pneumonia and atelectasis. However, the operative times were significantly longer with robotic lung resection. (4) Conclusions: The present meta-analysis demonstrated superior post-operative morbidity and mortality outcomes with robotic lung resection compared to open thoracotomy for non-small cell lung cancer.

## 1. Introduction

Conventional open thoracotomy has been the accepted standard treatment for resectable non-small cell lung cancer (NSCLC). However, the introduction of minimally invasive techniques in recent decades has revolutionized thoracic surgery. Video-assisted thoracoscopic surgery (VATS) has demonstrated superior efficacy and postoperative outcomes compared to open thoracotomy in a previous meta-analysis [1]. More recently, robotic video-assisted thoracoscopic surgery (RVATS) has emerged as a feasible alternative by offering three-dimensional visualization, enhanced precision and ergonomic advantages [2].

Several meta-analyses comparing RVATS, VATS and open resection have demonstrated similar safety and efficacy outcomes [3,4]. However, the literature directly comparing the peri-operative outcomes and complications of RVATS and open thoracotomy is scarce [5]. Therefore, this systematic review and meta-analysis is aimed to address this knowledge gap and compare the perioperative outcomes between robotic and open resection for NSCLC.

## 2. Materials and Methods

### 2.1. Literature Search Strategy

A systematic review was performed, using online databases including PubMed, EMBASE and the Cochrane Database of Systematic Reviews from their dates of inception to April 2022. The search terms included (robot * or “robotic assisted surgery” or “da Vinci”) and (open or thoracotom * or lobectom * or segmentectom * or pneumonectomy) and (“lung cancer” or “lung neoplasm” or “NSCLC”) as either keywords or Medical Subject Headings. The reference lists of all of the retrieved articles were reviewed for additional potentially relevant studies.

### 2.2. Selection Criteria and Data Extraction

The selected studies included those that compared patients with histologically proven NSCLC who underwent pulmonary resection by RVATS or open thoracotomy. The publications were limited to human subjects and English language. The exclusion criteria included studies with ten or fewer patients, aggregate data combining VATS and RVATS, case reports, conference abstracts, posters, editorials, systematic reviews or meta-analyses. For the studies published from the same institution using the same repeated population over time, only the most recent data were included for quantitative appraisal. Data were extracted from the article text, tables, figures and Appendix A.

### 2.3. Statistical Analysis

A random-effects meta-analysis of proportions or means was performed for pooling of categorical or continuous variables. The pooled data are presented as N (%) with 95% confidence intervals (CI). For the analysis of continuous data, the data presented as median and IQR were converted to mean and standard deviation, using the method by Wan [6]. Dichotomous or continuous variables were compared by using odds ratios (OR) or standard mean difference (SMD), respectively. I^2^ statistic was used to estimate the percentage of total variation across the studies due to heterogeneity, rather than chance. The thresholds for I^2^ values for low, moderate and high heterogeneity were considered as 0–49%, 50–74% and ≥75%, respectively. Two-sided *p* values less than or equal to 0.05 were considered statistically significant. All of the statistical analyses were conducted with Review Manager Version 5.4 (Cochrane Collaboration, Software Update, Oxford, UK).

## 3. Results

### 3.1. Quantity of Studies

A total of 1186 articles were identified through the electronic search. Exclusion of the duplicate studies yielded 910 potentially relevant articles for screening. Following review of the title and abstract, 863 studies were excluded and a full text review was performed on the remaining 47 articles. Twenty-two studies reporting on a total of 104,472 patients who underwent either RVATS (*n* = 12,061) or open thoracotomy (*n* = 92,411) lung resection met the selection criteria [7,8,9,10,11,12,13,14,15,16,17,18,19,20,21,22,23,24,25,26,27,28]. A summary of the study selection process is shown in Figure 1 as a PRISMA flowchart.

### 3.2. Quality of Studies

From the included studies, there was one randomized controlled trial (RCT) and twenty-one retrospective observational studies, of which eleven were propensity matched. Five of the studies compared only RVATS versus open thoracotomy and the remaining sixteen studies compared RVATS versus VATS versus open thoracotomy. A summary of the study characteristics is presented in Table 1.

### 3.3. Patient Characteristics

The overall median age was 67 (IQR 65–68) for RVATS and 66 (IQR 63–67) for open resection. The median percentage of males was 47% (IQR 44–53%) and 50% (IQR 48–60%) in the RVATS and open resection groups, respectively. The majority of the patients had a preoperative histopathological diagnosis of adenocarcinoma, comprising, on average, 68% of the RVATS patients and 59% of the open thoracotomy patients. This was followed by squamous cell carcinoma, seen in 25% of the RVATS group and 32% of the open thoracotomy group. Clinical staging was reported according to the seventh or eighth edition of the TNM staging system, with most of the patients classified as clinical Stage I or II. Median tumor size was 3 cm (IQR 2.5–3.2 cm) in the RVATS group and 3.2 cm (IQR 3.1–3.5) in the open thoracotomy group. Further baseline preoperative characteristics of patients are presented in Table 2.

### 3.4. Intra-Operative Outcomes and Surgical Approach

The operative time was significantly longer for the RVATS procedures compared to the open resection (SMD = 0.38, 95% CI 0.13–0.63, *p* = 0.003, I^2^ = 97%), based on the analysis of fourteen studies. The number of lymph node stations harvested was reported in five studies, and shown to be significantly higher using the RVATS approach compared to open thoracotomy (SMD = 0.62, 95% CI 0.46–0.78, *p* < 0.001, I^2^ = 49%). The rate of open conversion for RVATS was reported in eight studies, with an IQR of 6–9%. A summary of the intra-operative and surgical details is presented in Table 3.

### 3.5. Post-Operative Morbidity and Mortality Outcomes

Mortality was defined as death within 30 days or death within the same admission for all of the selected studies. Pooled analysis of fifteen studies demonstrated significantly lower mortality rates amongst the patients who underwent RVATS compared to open lung resection (OR 0.65, 95% CI 0.43–0.99, *p* = 0.04, I^2^ = 31%), Figure 2. In addition, the patients who underwent RVATS had a significantly shorter length of hospital stay compared to patients receiving open thoracotomy (SMD= −0.53, 95% CI −0.74 to −0.32, *p* < 0.001, I^2^ = 98%). There was no significant difference between the rates of reoperation between the two surgical approaches (OR 0.82, 95% CI 0.54–1.25, *p* = 0.35, I^2^ = 38%). Overall, the postoperative complications were significantly lower in the RVATS group compared to open surgery (OR 0.68, 95% CI 0.64–0.74, *p* < 0.001, I^2^ = 0%), based on the analysis of eleven studies in Figure 3. The most common complications were pneumonia, prolonged air leak, atelectasis, atrial arrythmia and post-operative bleeding in both of the groups. A meta-analysis of the pooled data for these outcomes showed significantly lower incidences of post-operative transfusion requirements (OR 0.37, 95% CI 0.2–0.66, *p* < 0.001, I^2^ = 71%), pneumonia (OR 0.61, 95% CI 0.46–0.81, *p* < 0.001, I^2^ = 28%) and atelectasis (OR 0.56, 95% CI 0.38–0.83, *p* < 0.001, I^2^ = 56%) for RVATS compared to open thoracotomy. The chest drain-duration was significantly shorter for RVATS (SMD = 0.048, 95% CI −0.78 to −0.17, *p* = 0.002, I^2^ = 95%), however the incidence of prolonged air leak was not significantly different between the two groups (OR 0.8, 95% CI 0.54–1.17, *p* = 0.25, I^2^ = 86%). Atrial arrhythmia was reported in thirteen studies and a pooled analysis showed a significantly lower incidence in RVATS patients (OR 0.76, 95% CI 0.69–0.83, *p* < 0.001, I^2^ = 0%), as seen in Figure 4. There was no significant difference in the rate of cardiovascular complications, including myocardial infarction (OR 0.84, 95% CI 0.53–1.34, *p* = 0.46, I^2^ = 0%) or thromboembolism (OR 0.59, 95% CI 0.28–1.26, *p* = 0.17, I^2^ = 35%) between the groups. In addition, the rates of wound infection were comparable between both of the groups (OR 0.71, 95% CI 0.37–1.36, *p* = 0.30, I^2^ = 47%). The details of post-operative outcomes are summarized in Table 4, and in the forest plots in Appendix A.

## 4. Discussion

Since the first publication on robotic thoracic surgery in 2002, the feasibility of this novel minimally invasive surgery in the treatment of NSCLC has been well demonstrated [29]. While VATS has demonstrated comparable safety and oncological outcomes compared to open resection [30], technical constraints, such as limited range of movement and poor ergonomics, may limit effective anatomic resection and lymph node dissection. Robotic platforms have offered distinct advantages in allowing up to seven degrees of freedom, dexterity, improved 3D visualization and magnification, as well as greater precision and stability [2]. These advantages have provided thoracic surgeons with an alternative surgical approach to mediastinal lesions, segmentectomies and sleeve resections that require complex dissection or suturing [31]. However, the potential disadvantages of higher costs and longer operating times associated with RVATS have been previously acknowledged [13,14,16]. Randomized controlled data comparing minimally invasive and open techniques have been relatively lacking, and the optimal approach is unclear.

The present meta-analysis aimed to provide an overview of the peri-operative outcomes of RVATS compared with open resection in patients with NSCLC. The key findings included a significantly lower rate of post-operative morbidity and mortality with the RVATS approach compared to open thoracotomy. The patients who underwent RVATS had significantly shorter lengths of hospital stay, shorter chest drain duration, fewer pulmonary complications and blood-product requirements after surgery. In addition, there were significantly lower rates of atrial arrhythmia observed in the RVATS group. In terms of lymph node management, a significantly higher number of the lymph-node stations were harvested through the RVATS approach. However, operation times were significantly longer for RVATS compared to open resection.

The superior outcomes of RVATS over open thoracotomy observed from this meta-analysis corroborate with previous reports. In an earlier meta-analysis directly comparing RVATS to open thoracotomy, Zhang [5] showed significantly reduced perioperative mortality and overall morbidity rates in the RVATS group. However, their study notably predates a growing number of studies published in more recent years and does not compare individual postoperative complications. A more recent network meta-analysis of 34 studies of 183,426 patients by Aiolfi [3] showed significantly reduced 30-day mortality, pulmonary and overall complications, as well as equivocal oncological and five-year survival outcomes in RVATS compared to VATS and open thoracotomy. However, a multicenter randomized controlled trial by Huang [11], comparing RVATS with open thoracotomy in 148 patients with N2 NSCLC, reported only significant reductions in blood loss, pain and chest drain duration in the RVATS group, but no difference in the other postoperative outcomes. In contrast to the previous studies, this present meta-analysis has shown a significant reduction in atrial arrhythmia and a significant increase in the number of lymph nodes stations harvested in the RVATS group compared to open thoracotomy. This may be attributed to the dexterity advantages of robotic instruments in tight spaces, that have been previously hypothesized to contribute to a higher rate of lymph node dissection and reduced tissue trauma [9,20]. A key disadvantage of RVATS has been its lengthier operative times, a finding reflected in this meta-analysis. The proponents of the robotic platform have suggested that this may partially reflect the initial learning curve, and several studies have shown reductions in operative times with increased volume over time [18,32].

Several limitations should be acknowledged in this study and the results should therefore be interpreted with caution. One limitation was a lack of standardized definitions of the endpoints between studies, such as the reporting of operation time, reasons for conversion and other morbidity outcomes. Ideally, the complications would be reported according to standardized criteria, such as the Clavien–Dindo classification, but this was inconsistently defined by individual studies, so a meta-analysis of the major complications was not possible. Variations in the patient inclusion criteria, baseline characteristics, resection type, neoadjuvant therapy, center volume, type of robotic model used and surgeon expertise may also have impacted on the outcome data. Another limitation was the inherent lack of randomized controlled data and a high proportion of studies that did not have propensity matching. Furthermore, statistical limitations included a relatively high degree of heterogeneity identified among the studies and a potential overlapping of the patients between the databases used in different studies.

Overall, the results of this meta-analysis reaffirmed the feasibility and safety of the RVATS approach. Robotic resections demonstrated significantly superior perioperative outcomes compared to open thoracotomy. This study also identified a lower incidence of atrial arrhythmias and a higher number of lymph node stations harvested using the RVATS approach, which has not been identified in previous meta-analyses. Randomized controlled data with well-defined surgical outcomes are needed in future to support these findings. Further innovation of the robotic platform and improved accessibility and affordability will help consolidate its role in the surgical management of lung cancer.

## Figures and Tables

**Figure 1 jpm-12-01311-f001:**
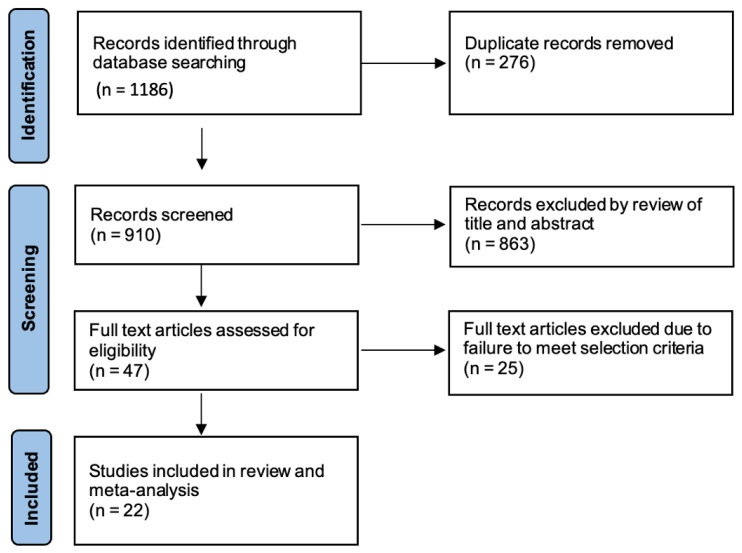
PRISMA flowchart detailing literature search process for studies comparing RVATS and open thoracotomy for non-small cell lung cancer.

**Figure 2 jpm-12-01311-f002:**
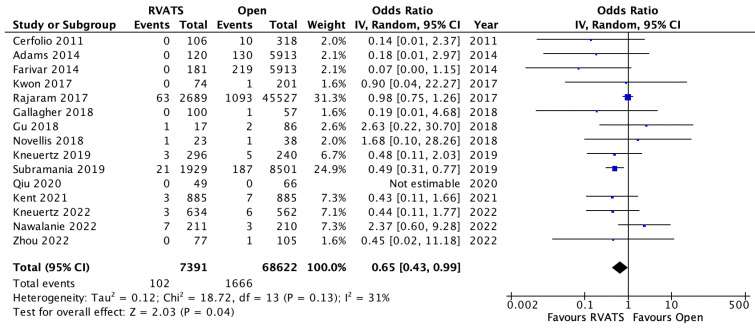
Mortality outcomes in patients who underwent robotic video-assisted thoracoscopic surgery compared with open thoracotomy for resectable non-small cell lung cancer [7,8,9,10,12,13,14,16,17,18,20,22,23,26,27].

**Figure 3 jpm-12-01311-f003:**
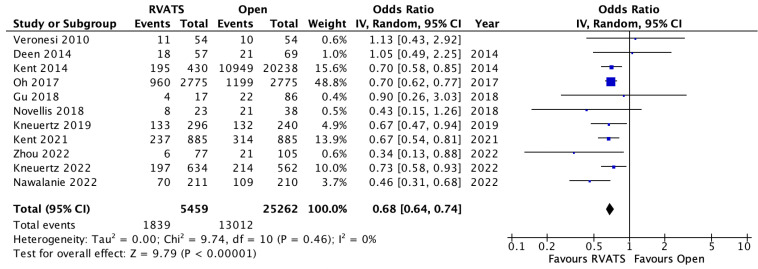
Overall complications in patients who underwent robotic video-assisted thoracoscopic surgery compared with open thoracotomy for resectable non-small cell lung cancer [8,9,10,13,16,17,19,24,25,28].

**Figure 4 jpm-12-01311-f004:**
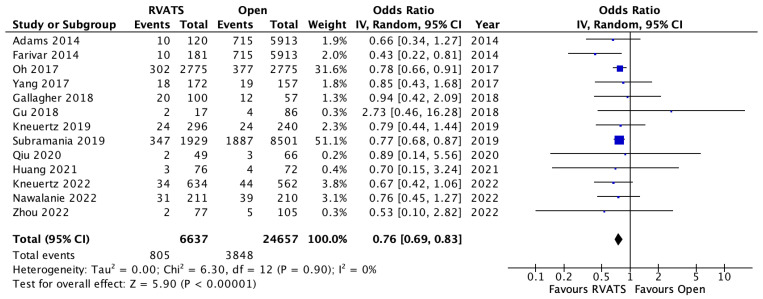
Atrial arrythmia in patients who underwent robotic video-assisted thoracoscopic surgery compared with open thoracotomy for resectable non-small cell lung cancer [7,8,9,11,12,13,14,17,18,19,21,23,26].

**Table 1 jpm-12-01311-t001:** Characteristics of studies comparing robotic video-assisted thoracoscopic surgery with open thoracotomy for patients with resectable non-small cell lung cancer.

Author	Year	Country	Period	Number of Participants RVATS Thoracotomy		Follow-Up Months (RVATS/Open)
Kneuertz [7]	2022	USA	2012–2017	634	562	NR
Nawalanie [8]	2022	USA	2006–2016	211	210	NR
Zhou [9]	2022	USA	2015–2019	77	105	25/25
Kent [10]	2021	USA	2013–2019	885	885	NR
Huang [11]	2021	China	2016–2020	76	72	24/24
Qiu [12]	2020	China	2012–2017	49	66	21/24
Kneuertz [13]	2019	USA	2012–2017	296	240	NR
Subramania [14]	2019	USA	2008–2014	1929	8501	NR
Nelson [15]	2019	USA	2011–2017	106	424	27/27
Novellis [16]	2018	Italy	2015–2016	23	38	NR
Gu [17]	2018	China	2014–2015	17	86	20/20
Gallagher [18]	2018	USA	2007–2014	100	57	NR
Oh [19]	2017	USA	2011–2015	2775	2775	NR
Kwon [20]	2017	USA	2010–2014	74	201	24/28
Yang [21]	2017	China	2002–2012	172	157	NR
Rajaram [22]	2017	USA	2010–2012	3689	45,527	NR
Farivar [23]	2014	USA	2010–2012	181	5913	NR
Kent [24]	2014	USA	2008–2010	430	20,238	NR
Deen [25]	2014	USA	2008–2012	57	69	NR
Adams [26]	2014	USA	2010–2012	120	5913	NR
Cerfolio [27]	2011	USA	2010–2011	106	318	NR
Vernoesi [28]	2010	Italy	2006–2008	54	54	NR

RVATS = robotic video-assisted thoracoscopic surgery; NR = Not reported.

**Table 2 jpm-12-01311-t002:** Summary of baseline characteristics of patients who underwent robotic video-assisted thoracoscopic surgery compared with open thoracotomy for resectable non-small cell lung cancer.

Author	Age (Years)		Male (%)		BMI		FEV1 (%)		DLCO (%)		TNM Clinical Staging (%) (I/II/III+)		Histopathology (%) (ADC/SCC/Other)		Tumor Size (cm)	
	RVATS	Open	RVATS	Open	RVATS	Open	RVATS	Open	RVATS	Open	RVATS	Open	RVATS	Open	RVATS	Open
Kneuertz [7]	69	68	41	43	28	28	79	79	71	70	93/7/0	90/10/0	73/19/8	72/22/6	NR	NR
Nawalanie [8]	65	62	46	52	27	26	86	78	83	73	NR	NR	NR	NR	NR	NR
Zhou [9]	65 *	59 *	51	53	28 *	29 *	94	92	95	85	NR	NR	NR	NR	1.7	2.3
Kent [10]	67	67	46	49	28	28	87	86	NR	NR	71/20/9	69/24/8	74/17/9	61/32/7	3.1	3.4
Huang [11]	61	61	67	71	NR	NR	89	90	94	90	36/32/37	29/24/47	NR	NR	3.3	3.6
Qiu [12]	61	61	89	91	24	24	NR	NR	NR	NR	NR	NR	NR	NR	NR	NR
Kneuertz [13]	64 *	64 *	57	50	28 *	28 *	81 *	83 *	77 *	77 *	NR	NR	NR	NR	NR	NR
Subramania [14]	69	68	44	49	NR	NR	NR	NR	NR	NR	NR	NR	NR	NR	NR	NR
Nelson [15]	67	66	44	50	NR	NR	86	86	NR	NR	74/16/10	49/26/24	75/25/0	74/26/0	3	3.2
Novellis [16]	70 *	71 *	NR	NR	NR	NR	90*	90*	NR	NR	52/26/22	69/17/14	NR	NR	2.1 *	3 *
Gu [17]	62	61	100	93	23	24	75	81	84	85	35/30/35	42/34/24	0/76/18	11/72/13	3.5	3.6
Gallagher [18]	68 *	66 *	98	96	NR	NR	76 *	72 *	73 *	73 *	84/16/0	72/28/0	NR	NR	NR	NR
Oh [19]	67	67	47	47	NR	NR	NR	NR	NR	NR	NR	NR	NR	NR	NR	NR
Kwon [20]	67 *	66 *	38	56	NR	NR	NR	NR	NR	NR	NR	NR	NR	NR	NR	NR
Yang [21]	68	68	43	34	NR	NR	92	90	85	83	100/0/0	100/0/0	11/53/13	14/46/14	NR	NR
Rajaram [22]	68	67	45	48	NR	NR	NR	NR	NR	NR	NR	NR	62/24/14	58/28/14	NR	NR
Farivar [23]	65	65	42	50	28	28	84	80	74	74	NR	NR	NR	NR	NR	NR
Kent [24]	67	66	44	49	NR	NR	NR	NR	NR	NR	NR	NR	NR	NR	NR	NR
Deen [25]	68	68	50	87	NR	NR	87	87	80	77	42/9/3	48/12/9	NR	NR	2.8	3.2
Adams [26]	65	65	48	50	27	28	79	80	73	74	NR	NR	NR	NR	NR	NR
Cerfolio [27]	66 *	66 *	48	47	NR	NR	84	85	76	80	NR	NR	NR	NR	3.7 *	3.6 *
Vernoesi [28]	NR	NR	38	34	NR	NR	95	95	NR	NR	45/5/4	42/4/8	NR	NR	NR	NR

* = median value, all other values are reported as mean; NR = not reported; BMI = body mass index; FEV1 = predicted forced expiratory volume in 1 s; DLCO = diffusion lung capacity for carbon monoxide; ADC = adenocarcinoma; SCC = squamous cell carcinoma; TNM = TNM classification of malignant tumors.

**Table 3 jpm-12-01311-t003:** Summary of intraoperative outcomes of patients who underwent robotic video-assisted thoracoscopic surgery compared with open thoracotomy for resectable non-small cell lung cancer.

Author	Resection Type	Operation Time (Mins)		Lymph Nodes Harvested		Stations Harvested		Conversion to Open (%)
		RVATS	Open	RVATS	Open	RVATS	Open	RVATS
Kneuertz [7]	S	239	227	10	8	5	4	NR
Nawalanie [8]	P, L, B, S	150 *	160 *	23	13	5	3	NR
Zhou [9]	S	205	147	14	10	6	4	0
Kent [10]	L	166	164	NR	NR	NR	NR	NR
Huang [11]	L	104	102	NR	NR	NR	NR	NR
Qiu [12]	B	200	240	23	23	NR	NR	0
Kneuertz [13]	L	287 *	279 *	NR	NR	NR	NR	NR
Subramania [14]	L	NR	NR	NR	NR	NR	NR	NR
Nelson [15]	L	226 *	148 *	17 *	12 *	6	5	8
Novellis [16]	L	155	122	NR	NR	5	4	9
Gu [17]	B	155	150	NR	NR	NR	NR	6
Gallagher [18]	L	195 *	175 *	NR	NR	5	4	NR
Oh [19]	L	276	235	NR	NR	NR	NR	7
Kwon [20]	L, S	233	268	NR	NR	NR	NR	19
Yang [21]	L	NR	NR	NR	NR	NR	NR	19
Rajaram [22]	L	NR	NR	NR	NR	NR	NR	NR
Farivar [23]	L, S	199	244	NR	NR	NR	NR	NR
Kent [24]	L	NR	NR	NR	NR	NR	NR	NR
Deen [25]	L, S	223	180	NR	NR	NR	NR	NR
Adams [26]	L	242	176	NR	NR	NR	NR	NR
Cerfolio [27]	L	132	90	17	15	8	8	NR
Vernoesi [28]	L	235 *	154 *	17	18	4	7	9

* = median value, all other values without asterisks are reported as mean; NR = not reported; P = pneumonectomy, L = lobectomy, B = bronchial sleeve, S = segmentectomy.

**Table 4 jpm-12-01311-t004:** Summary of postoperative outcomes of patients who underwent robotic video-assisted thoracoscopic surgery compared with open thoracotomy for resectable non-small cell lung cancer.

Study	Mortality (%)		Overall Complications (%)		Length of Stay (Days)		Chest Drain Duration (Days)		Post-Operative Transfusion (%)		Pneumonia (%)		Prolonged Air Leak (%)		Atelectasis (%)		Atrial Arrhythmia (%)	
	RVATS	Open	RVATS	Open	RVATS	Open	RVATS	Open	RVATS	Open	RVATS	Open	RVATS	Open	RVATS	Open	RVATS	Open
Kneuertz [7]	0.5	1.1	31	38	4	5	NR	NR	3	4	4	3.7	8.1	7	2.6	3	5.4	7.8
Nawalanie [8]	3.3	1.6	33	52	3	5	NR	NR	NR	NR	7.6	11.9	7.6	20.3	6.1	13	14.7	18.6
Zhou [9]	0	1	8	20	3	4	2	3	1	6	NR	NR	3.9	13.3	NR	NR	2.6	4.8
Kent [10]	0.3	0.8	27	36	4 *	6 *	4	5	4	5	NR	NR	NR	NR	NR	NR	NR	NR
Huang [11]	NR	NR	NR	NR	10 *	11 *	4 *	5 *	4 *	5 *	3.9	8.3	7.9	8.3	NR	NR	3.9	5.6
Qiu [12]	0	0	NR	NR	NR	NR	NR	NR	NR	NR	5	4.6	6.8	3.1	5	5.9	4.4	4
Kneuertz [13]	1	2	45	55	4 *	5 *	NR	NR	NR	NR	3	8	5	9	5	16	8	10
Subramania [14]	NR	NR	NR	NR	4 *	7 *	NR	NR	NR	NR	5.2	10.1	8	3.8	NR	NR	18	22.2
Nelson [15]	NR	NR	NR	NR	4 *	5 *	NR	NR	NR	NR	7	4	15	16	NR	NR	NR	NR
Novellis [16]	4.4	2.6	35	53	4 *	6 *	NR	NR	NR	NR	NR	NR	NR	NR	NR	NR	NR	NR
Gu [17]	6	2	24	26	11	10	9	7	9	7	12	9	NR	NR	NR	NR	12	5
Gallagher [18]	0	1.8	NR	NR	6 *	10 *	3 *	6 *	3 *	6 *	12	14	NR	NR	NR	NR	20	21
Oh [19]	NR	NR	35	43	7	9	NR	NR	NR	NR	NR	NR	10.1	9	12.4	15.7	10.9	13.6
Kwon [20]	0	0.5	NR	NR	4 *	6 *	3 *	4 *	3 *	4 *	NR	NR	2.7	10	NR	NR	NR	NR
Yang [21]	0	0	NR	NR	4 *	5 *	NR	NR	NR	NR	2.9	5	8.7	4.5	2.9	2.5	10.5	12.1
Rajaram [22]	1.7	2.4	NR	NR	6	7	NR	NR	NR	NR	NR	NR	NR	NR	NR	NR	NR	NR
Farivar [23]	0	2	NR	NR	3	7	3	5	3	5	1.7	5.1	6.1	10.7	1.7	5.3	5.5	12.1
Kent [24]	NR	NR	45	54	6 *	8 *	NR	NR	NR	NR	NR	NR	NR	NR	NR	NR	NR	NR
Deen [25]	NR	NR	32	30	5	6	NR	NR	NR	NR	NR	NR	0	0	NR	NR	NR	NR
Adams [26]	0	2.2	NR	NR	5	7	3	5	3	5	1.7	5.1	5.2	10.8	NR	NR	8.6	12.1
Cerfolio [27]	0	3	NR	NR	2 *	4 *	2 *	3 *	2 *	3 *	NR	NR	NR	NR	NR	NR	NR	NR
Vernoesi [28]	NR	NR	20	19	4 *	6 *	NR	NR	NR	NR	NR	NR	NR	NR	NR	NR	NR	NR

* = median value, all other values are reported as mean; Mortality = mortality in hospital or within 30 days; Prolonged air leak ≥ 5 days; NR = not reported.

## Data Availability

The data presented in this study are available in the article and Appendix A.

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
