# Peer review of "Complications of Robotic Video-Assisted Thoracoscopic Surgery Compared to Open Thoracotomy for Resectable Non-Small Cell Lung Cancer"

_jpm, 2022, doi:10.3390/jpm12081311_

Round 1

Reviewer 1 Report

This paper represents a meta-analysis examining the differences and outcomes in pulmonary lobectomy between robotic VATS and open thoracotomy in the treatment of NSCLC.  A total of 22 studies were selected that comprised a total of 104,472 patients, of which roughly 12k underwent RVATS and 92k had a traditional thoracotomy.   Despite this imbalance, significant findings were found with improvements in morbidity, length of stay, chest tube duration, and lymph node harvest favoring the RVATS group and only operative time favoring the open group.  

The overall analysis is well done and the conclusions support what is already known in the literature in that minimally invasive thoracic surgery has overall better outcomes when compared to open thoracotomy.   The following are my suggestions/questions:

1) Given that 16/22 studies also had data on traditional VATS and robotic VATS, could the authors include a comparison between traditional VATS and robotic VATS?   This may be the more pressing topic today in the field of thoracic surgery as many argue the pros and cons for each.

2) The studies included span an almost 12 year period of time (2010-2022).  In that period, the robotic instrumentation (likely DaVinci) has undergone significant upgrade/re-design including increasing from 3 to 4 arms.   Do the authors find any significant differences in the RVATS outcomes from the earlier robotics studies when compared to the later group?

Author Response

Please see the attachment, thank you kindly.

Reviewer 2 Report

The authors brilliantly reviewed the real-world evidence of robotic resection versus thoracotomy for resectable non-small cell lung cancer, and concluded that robotic this meta-analysis reaffirmed the feasibility and safety of the RVATS approach. Further, robotic resections demonstrated significantly superior perioperative outcomes compared to open thoracotomy.

The following are my comments.

1. As you diligently explained the possible limitation in the current study. I would still like to know whether any included data discussing the issue of component of GGO in stage I lung cancer, comparison of hospital charge, different robotic system (DaVinci Xi or Si) and so on.

2. As your study focused on complications. The major and minor parts should be clarified. For example, Clavien-Dindo classification should be used. Thanks.

Author Response

(The authors gave the same response as above.)
